**Factors influencing development of cracking-sliding failures of loess across the eastern Loess Plateau of China**

Yanrong Li[1], Jiarui Mao[2], Xiqiong Xiang[2], Ping Mo[1]

[1]Department of Earth Sciences and Engineering, Taiyuan University of Technology, Taiyuan, 030024, China

[2]Guizhou University, Guiyang, 550025, China

*Correspondence to:* Yanrong Li (li.dennis@hotmail.com) & Xiqiong Xiang (tujia@126.com)

**Abstract:** Loess is a porous, weakly cemented, and unsaturated Quaternary sediment deposited by the wind in arid and semiarid regions. Loess is widely and thickly distributed in China, making the Loess Plateau the largest bulk accumulation of loess on Earth. However, the fragile geoenvironment in the loess areas of China causes frequent and various geohazards, such as cracking–sliding failure ("beng-hua" in Chinese), which is a typical geohazard that causes the largest number of casualties each year. This study investigates the main influencing factors and development patterns of cracking–sliding failure of loess to help prevent its occurrence and reduce losses effectively. The following conclusions are derived: (1) cracking–sliding failures mostly take place in rectilinear slopes, convex slopes, slopes with gradients greater than 60°, slopes with heights of 5 m to 40 m, and slopes mostly exposed to sunlight with aspects of 180° to 270°; (2) cracking–sliding failures occur mostly from 10 pm to 4 am and mainly in the rainy season (July to September) and in the freeze–thaw season (March to April); and (3) highly intense human activities in the region correspond to a high possibility of cracking–sliding failures.

**Keywords:** loess, cracking–sliding failure, influencing factors, development patterns

## 1 Introduction

Any yellowish, carbonate-bearing, quartz-rich, silt-dominated strata formed by aeolian deposition and aggregated by loessification during glacial times are widely defined as loess (Richthofen, 1882; Pye, 1987). Loess ("huang-tu" in Chinese) and its related deposits are one of the most widespread Quaternary sedimentary formations, and they are most abundant in arid or semiarid regions in inner Eurasia and North America; they are characterized by high porosity, weak cementation, and unsaturation (Samlley et al., 2011).

The Loess Plateau in China (LPC) is the main region for comprehensive development of agriculture, forestry, animal husbandry, and industrial resources with an arable land area of 173,000 km$^2$, which accounts for more than one-fifth of the entire arable land of the country and

feed more than 200 million people (Zhang, 2014). However, geohazards, such as cracking–sliding, toppling, falling, sliding, peeling, and failures of cavity built by human, occur frequently because of fragile geological and natural environments, excessive reclamation, and unreasonable engineering activities. Among these geohazards, cracking–sliding failure, normally with a volume of several hundred cubic meters, causes the largest number of casualties in the eastern LPC (Lei, 2001). More than 1000 cracking–sliding failures were recorded in the past two decades, and they caused an average of more than 100 fatalities per year despite the small volumes of individual failures. Unlike "flows" or "slides" as defined by Cruden and Varnes (1996), cracking–sliding failures have composite failure planes composed of two parts. The upper part normally develops vertically from the crown of the slope down for one to several meters deep. The upper part forms by tensile cracking, but the slope can stand stably for a long time with such cracks. The lower part is generally inclined at an angle ranging from 15° to 60°. Sliding along the lower part, which is triggered by rainfall, freezing–thawing, daily temperature fluctuations, slope undercutting, and earth tremors likely mobilizes cracking–sliding failures.

According to historical records, 62 cracking–sliding failures occurred in Shenmu, Mizhi, Zizhou, and other sites in Northern Shaanxi Province from 1985 to 1993 and caused 258 deaths and more than 40 injuries (Qu et al., 2001). In 2005, the cracking–sliding failure in Jixian County in Shanxi Province resulted in 24 deaths and economic losses of approximately RMB 10 million. Failure with a volume of $2.5 \times 10^4$ m$^3$ took place in Zhongyang County in Shanxi on November 16, 2009, causing 23 deaths and destroying 6 houses. In 2013, 36 loess failures were documented in Tianshui City, Gansu Province (Xin et al., 2013). In 2015, a cracking–sliding failure in Linxian County, Shanxi buried four families comprising nine people. All of these failures developed within the loess–paleosol sequence, with relatively uniform mineralogical and chemical compositions. More recently, a cracking-sliding failure occurred in Shilou County of Shanxi Province on March 10, 2018, and destroyed 36 houses (Fig. 1). The original loess slope was characterized by slope gradient of 60°, height of 50 m and aspect of 280°. The scarp of this failure was dominated by near-vertical tensile cracks with average gradient of 85°. The displaced mass reached 7600 m$^3$.

Frequent and disastrous events demand an in-depth understanding of causative factors and development patterns of loess failures to reduce the occurrence of such geohazards. This study collects a large set of data on loess cracking–sliding failures, climate, and soil temperature to facilitate a detailed analysis of the internal and external causes of such failures. This study also emphasizes the influences of slope features (i.e., slope type, gradient, height, and aspect), rainfall, freezing–thawing cycles, daily temperature fluctuations, and human engineering activities.

**2 Study area**

The study area is limited to the eastern LPC and covers Northern Shaanxi and Western Shanxi provinces, considering the homogeneous background of climatic, morphologic, geologic, and anthropic conditions in these regions (Fig. 2). Elevation of the study area ranges from 800 to 1300 m above sea level from southeast to northwest. The study area has a typical semiarid continental monsoon climate with four distinct seasons. The average annual rainfall in this area varies from 400 to 700 mm. Rainfall in summer (from July to September) accounts for approximately 70% of the year (Hui, 2010; Qian, 2011; Zhu 2014). For instance, the maximum precipitation in an hour in Yan'an City can accumulate to more than 60 mm in summer (Zhu, 2014). The total rainfall in Shilou County reached 412 mm in a month from early July to early August in 2013 (Lv, 2011) and corresponded to 81% of rainfall in the same year. According to records for the past 10 years, the average annual temperature is relatively constant, ranging from 8 °C to 12 °C. However, variations in temperature in a day can occasionally be higher than 25 °C, the highest temperature being recorded at noon and the lowest temperature at midnight.

The study area is located in the east of the Ordos basin. The Fenwei Graben, spanning northeast to southwest, is a tectonic depression encountering a number of normal and strike–slip faults and covering more than 20,000 km$^2$ (Huang et al., 2008; Liu et al., 2013). The thickly bedded Pleistocene loess–paleosol sequence constitutes more than 70% of the study area and reaches a maximum thickness of 300 m. From top to bottom, the loess–paleosol sequence includes Late Pleistocene Malan Loess ($Q_3$), Middle Pleistocene Lishi Loess ($Q_2$), and Early Pleistocene Wucheng Loess ($Q_1$). The Malan Loess, with thickness ranging from 10 m to 30 m, is the most widespread. The Lishi Loess, with several to tens interlayers of loess and paleosol, underlies the Malan Loess and forms a 60–150 m thick layer. The Wucheng Loess is sporadically exposed along some loess gullies. Remarkable landforms, such as loess platforms, ridges, and hillocks, have been formed in the study area because of intensive surficial erosion (Zhang, 1983, 1986). Loess platforms are mainly distributed in the Luochuan area in Northern Shaanxi Province; loess ridges are mainly found in the peripheries of the Luochuan platform and the eastern regions of the Yellow River; and loess hillocks are mainly located in Yan'an, Suide and in both sides of the Yellow River between Shaanxi and Shanxi provinces.

**3 Dataset**

A large set of data of loess cracking–sliding failure events were collected from published literature and unpublished reports by local governments. Slope profile, gradient, height and aspect, were derived in polygon from the initiation areas. The initiation areas rather than the whole landslides were compared in the following statistical analysis. The polygons were obtained by means of 1) interpretation of remote sensing images which were taken prior to the event; 2) engineering documentation if the host slope was engineered; or 3) post-event field survey and consultation with the local population. The field survey was normally conducted within 1–2 days immediately after the event. A total of 1176 cracking–sliding events were

recorded in the past 20 years across the study area. Of these events, 321 were published in the
literature, 670 were presented in government reports, and 185 were unpublished. All of the 1176
failures were individually reviewed by verifying the reliability, accuracy, and completeness of
the original records. Finally, 458 cases (red dots in Fig. 2) were selected to set up the dataset for
this study by considering the completeness of the records of any type.

Data pertaining to rainfall were obtained from the records of 75 meteorological stations (blue
dots in Fig. 2), which are almost uniformly distributed across the study area. Statistical analysis
shows that the variation in average annual rainfall in the past 15 years among these stations is
less than 80 mm, indicating a relatively homogeneous climatic condition over the study area.

## 4 Results and discussion

### 4.1 Internal factors

Loess slopes are divided into four types in terms of slope profile: rectilinear, convex, concave,
and stepped slopes (Table 1). Concave and stepped slopes are more stable than the rectilinear and
convex slopes. We surveyed 212 loess slopes in Lishi City in Shanxi Province and found that
stepped slopes, convex slopes, rectilinear slopes, and concave slopes account for 38%, 31%, 18%,
and 13% of all slopes, respectively (Fig. 3a). This finding is consistent with the conclusion of
Qin et al. (2015), who performed a field survey on loess slopes in Yan'an City, Shaanxi Province.
However, approximately one-half of cracking–sliding failures occur in rectilinear slopes. In Fig.
3b, the statistical analysis of the 458 failure cases indicates that rectilinear slopes are the most
susceptible to cracking–sliding failure (48%), followed by convex slopes (28%). Stepped (13%)
and concave (11%) slopes are the least susceptible to such failures.

In general, the overall gradients of rectilinear and convex slopes are steep, resulting in large
internal stresses and stress concentrations, particularly at the shoulder and toe sections (Table 1).
The bottom part of the concave slope has a gentle gradient and has a supporting function to the
steep upper part, thereby relieving the stress concentration; the maximum shear stress at the foot
of concave slopes is typically only one-half of the shear stress at the foot of rectilinear slopes
(Zhang et al., 2009). The stress distribution pattern in each step section of a stepped slope is
similar to that of a rectilinear slope. However, the magnitude of internal stress of stepped slopes
is lower than that of rectilinear slopes because of the small height of each step and the gentle
overall gradient. These findings explain that most cracking–sliding failures occur in rectilinear
slopes, although these slopes are not the dominant slope type in the loess area.

In addition to slope profile, the gradient, height, and aspect of loess slopes are closely related
to the occurrence of cracking–sliding failures. Figure 4a shows that failure occurs mostly on
slopes with gradients greater than 60° and that the number of failures increases with gradients.
Of the cracking–sliding failures, 16%, 25%, and 47% occur on slopes with gradients ranging
from 61° to 70°, from 71° to 80°, and from 81° to 90°, respectively. Figure 5 shows the tension

zones that developed at slope shoulders, where radial and tangential stresses transform into
tensile stresses. The steeper the slope is, the wider the tension band is (Stacey, 1970; Zhang et al.,
2009).

Figure 4b illustrates that slope height is another main factor controlling the occurrence of
cracking–sliding failures. In the study area, most cracking–sliding failures occur on slopes with
heights of 5 m to 40 m and thus account for 87% of the total number of occurrences. The
remaining 13% take place on slopes with heights of more than 40 m. A high slope normally
develops a gentle gradient because of long-term weathering and erosion. By contrast, a low slope
generally forms a steep gradient (Zhu et al., 2011), thereby becoming prone to collapses.

Slopes mostly exposed to sunlight are more prone to the development of cracking–sliding
failures than shady slopes (Fig. 4c). Statistical analysis shows that 69% of the cracking–sliding
failures occur on slopes with aspects in the range of 90° to 270°, particularly within 180° to 270°
because sunward slopes receive long sunshine hours and soil temperature is relatively high
during the day. Therefore, high differences in temperature exist between day and night. Sunward
slopes are generally subjected to more weathering than shady slopes, resulting in fractured
structures, which are unfavorable to slope stability. Furthermore, people usually reside on
sunward slopes, and dense human engineering activities exert a large degree of disturbance on
the slope body, thereby increasing the occurrence of failures.

**4.2 External factors**

**1) Rainfall**

Rainfall remarkably influences the stability of loess slopes. In Figure 6, the number of loess
failures is closely and positively correlated with the average monthly rainfall of the past 15 years.
Summer rainfall (July to September) in the study area accounts for approximately 60% of the
annual precipitation, and the number of cracking–sliding failures in the same period corresponds
to 62% of the total failures. This finding is consistent with that of Gao et al. (2012), who
indicated that more than 60% of loess failures happen in Gansu Province in the rainy season.
Wei (1995) and Liu et al. (2012) presented a similar conclusion in Shanxi and Shaanxi provinces,
respectively.

Rainfall induces loess cracking–sliding failures in three ways, namely, splash erosion, shovel
runoff, and seepage. At the beginning of rainfall, soil particles with poor adhesion are separated
and broken under the impact of raindrops. When potholes formed by splash erosion are filled
with water, a layer of water flow forms and triggers small soil particles to move. Along with the
continued rain, this water flow converges into the slope runoff to erode and destroy the slope
surface further (Tang et al., 2015). In cases of persistent rainfall, preferential seepage pipes

usually develop inside a slope, thereby saturating the soils, reducing the shear strength, and
eventually causing cracking–sliding failures.

**2) Freezing and thawing**

Figure 6 shows that cracking–sliding failures occur frequently not only in the rainy season
from July to September but also in the winter-to-spring transition from March to April. Soil
temperature increases rapidly from values below 0°C to values above 0°C. As shown in Figure 7,
soil temperature remains negative, and the frozen depth can reach approximately 1.0 m
underground from late November to February in the loess areas in China. At the end of March,
the ground temperature begins to increase, and the frozen layer gradually enters the thawing
stage. By mid-April, the soil is rapidly heated up to approximately 8 °C.

Freezing and thawing mainly promote the occurrence of cracking–sliding failures via two
mechanisms: 1) Frost heaving damages the soil structure and reduces soil shear strength. The
loess itself contains a considerable number of large pores. Frost heaving further increases the
distance between soil particles, reduces the dry density of soil, and loosens the structure, thereby
reducing its cohesion and internal friction angle. 2) Thawing causes the loess structure to
collapse and reduce its shear strength. Thawed water can dissolve cement, especially calcareous
cement, between loess particles, consequently damaging the loess structure and increasing pore
water pressure; as a result, the shear strength of the soil decreases (Pang, 1986).

**3) Daily temperature fluctuation**

Consistent with previous findings (Wei, 1995), our results indicate a relatively high
frequency of occurrence of cracking–sliding failures between 10 pm and 4 am (Fig. 8). The
difference in temperature between day and night in the loess area is more obvious than that in
other regions at the same latitude in China (Sun and Zhang, 2011), and variations in air
temperature in a day can occasionally reach 30 °C. The soil at a 50 cm depth shows an average
daily temperature difference of approximately 5 °C in summer (Fig. 9). Thermal expansion and
shrinkage occur during the rapid change in day and night temperatures. Under the cyclic
functioning of shrinkage and expansion stresses, a soil structure loosens.

**4) Human activity**

Loess areas in China have a population of more than 200 million. Human engineering
activities frequently occur and mainly involve cutting slopes for buildings, excavation for cave
dwellings, and construction of terraced fields and roads (Del Prete and Parise, 2013). Cutting
slopes for buildings causes the side slope to become steep. Unloading-induced tensile fractures
are usually produced on the trailing edge of slopes during the rapid adjustment of a stress field
within a slope (Fig. 10a). When a cave is excavated, roof damage (normally caving) happens
because of a local tensile stress concentration if the design of a geometric section of a cave is

inappropriate (Fig. 10b). Terraced fields change the original path of surface runoff and enhance
rainfall infiltration. Together with irrigation, terraced fields increase the water content of loess
slopes and increase their phreatic level (Fig. 10c). The majority of traffic lines in the loess area
stretch along valleys and bank slopes. Slope cutting and excavation during road construction
result in a large number of high and steep side slopes, which provide a suitable environment for
failures (Fig. 10d).

More than half of the failures are attributed to human engineering activities (Fig. 11). In 2014,
9 of 16 failure cases that occurred in Yan'an City were caused by extremely steep slopes for cave
dwelling construction, and the 7 other cases were consequences of improper treatment of side
slopes for road construction (Lei, 2001). These findings demonstrate that intense human
activities likely result in a high probability of loess failures.

**5 Conclusions**

This study investigates the influencing factors and development patterns of loess cracking–
sliding failures in the eastern LPC according to a large collection of field investigation data. The
following conclusions are obtained.

(1) The influencing factors of cracking–sliding failures are divided into internal and
external causes. Internal causes include various features, such as slope geometry, height, gradient,
and aspect of loess slopes, whereas external causes comprise rainfall, freezing–thawing cycles,
temperature fluctuation, and human engineering activities.

(2) Cracking–sliding failure more likely occurs in rectilinear and convex slopes than in
concave and stepped slopes. Rectilinear and convex slope gradients are generally steep, stress
concentrations are obvious, and slope stability is poor. The stress concentration in concave and
stepped slopes is minimized, and stability is fair. Cracking–sliding failure more likely takes place
on slopes with gradients greater than 60°, and the greater the gradient is, the higher the likelihood
of failures. Cracking–sliding failure also tends to occur on slopes with heights of 5 to 40 m.
Slopes below 5 m have low internal stress and high stability. Slopes above 40 m are generally
gentle with low stress concentration. The dominant aspect of the development of cracking–
sliding failure is within 180° to 270° (sunward slopes) because of the evident temperature
difference between day and night and the strong weathering.

(3) The occurrence of cracking–sliding failure displays a particular time pattern. Within a
240 year, its occurrence coincides with seasonal rainfall. Failures mainly occur in the rainy season, or
241 from July to September. In addition, failures frequently take place from March to April because
of freezing and thawing. Within a day, failures happen mostly from 10 pm to 4 am because of the
large temperature variation between day and night.

(4) The more intense the engineering activities are, the greater the possibility of loess failures is. Human engineering activities in loess areas include cutting slopes for buildings, excavation of cave dwellings, and construction of terraced fields and roads. These engineering activities usually lead to a rapid change in the features and stress field of slopes. Such high and steep side slopes tend to develop unloading-induced tensile fractures, thereby increasing the possibility of loess failures.

*Acknowledgments*. This study was supported by the Key Program of National Natural Science Foundation of China (No. 41630640), the Major Program of the National Natural Science Foundation of China (No. 41790445), the 2014 Fund Program for the Scientific Activities of Selected Returned Overseas Professionals in Shanxi Province, Shanxi Scholarship Council of China, Outstanding Innovative Teams of Higher Learning Institutions of Shanxi, Soft-science Fund Project of Science and Technology in Shanxi, Research Project for Young Sanjin Scholarship of Shanxi, Collaborative Innovation Center for Geohazard Process and Prevention at Taiyuan Univ. of Tech., Recruitment Program for Young Professionals of China.

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

**Captions of figures and tables**

**Table 1.** Classification of loess slopes.

**Figure 1.** The cracking-sliding failure occurred in Shilou County of Shanxi Province on March 10, 2018 (110°50'48.54"E, 36°59'54.76"N).

**Figure 2.** Geological map of the study area. The red dots denote the cracking–sliding failure cases, and the blue dots indicate the meteorological stations in the study area.

**Figure 3.** Statistical analysis results: a) classification of loess slopes in Lishi City, Shanxi Province, China, on the basis of a field survey of 212 loess slopes, indicating that stepped slopes are dominant in the study area. b) Percentage of cracking–sliding failures that occurred in different types of loess slopes across the study area, showing that rectilinear slopes are highly susceptible to loess failures.

**Figure 4.** Effect of slope features on cracking–sliding failures.

**Figure 5.** Development of tension zones in slopes of different gradients (Stacey, 1970; Zhang et al., 2009).

**Figure 6.** Occurrence of cracking–sliding failures mainly in July to September and consistent with the average monthly rainfall.

**Figure 7.** Annual variation of temperature (°C) within a shallow zone of a typical loess slope (Yang and Shao, 1995).

**Figure 8.** Temporal distribution of cracking–sliding failures in a day between 10 pm and 4 am.

**Figure 9.** Daily soil temperature variation in loess areas of China in summer. Data are from the field monitoring during April 2014 to September 2017 in Linxian County, Shanxi, China.

**Figure 10.** Typical engineering activities in loess areas in China: a) cut slopes for buildings; b) excavations for cave dwellings; c) terraced fields for farming; and d) cut slopes for highways.

**Figure 11.** Role of engineering activities in loess failures: a) Shanxi Province; and b) Huangling County, Shaanxi Province.

**Table 1.** Classification of loess slopes.

| Slope type | Profile | Characteristics | Susceptibility to cracking–sliding failure |
|---|---|---|---|
| Rectilinear |  | Slope is straight or nearly straight; slope gradients are fairly large (>55°); stability is low. | Yes |
| Convex |  | Gentle at the top and steep at the bottom; convex shoulder; stability is generally poor. | Yes |
| Concave |  | Curves inward; gentle toward the toe supporting steep upper slope; more stable than other slopes; stability is fair. | No |
| Stepped |  | Stepped with straight faces; average gradient of the overall slope is generally small; stability is good. | No |

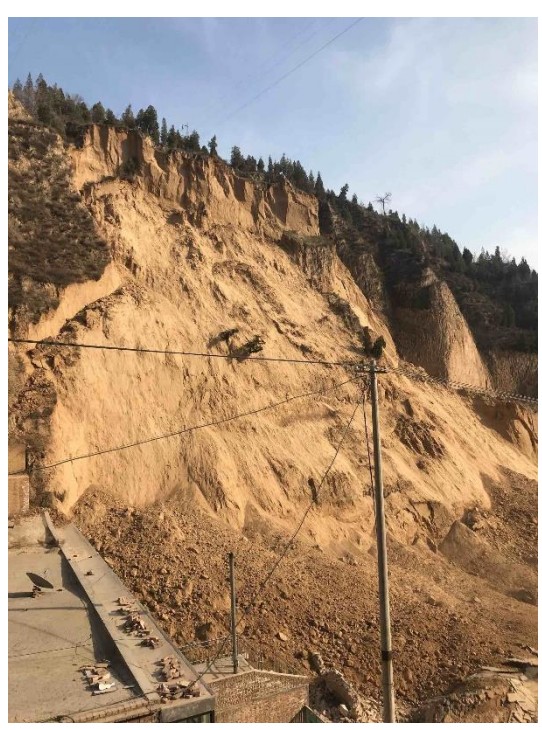

**Figure 1.** The cracking-sliding failure occurred in Shilou County of Shanxi Province on March 10, 2018 (110°50'48.54"E, 36°59'54.76"N).

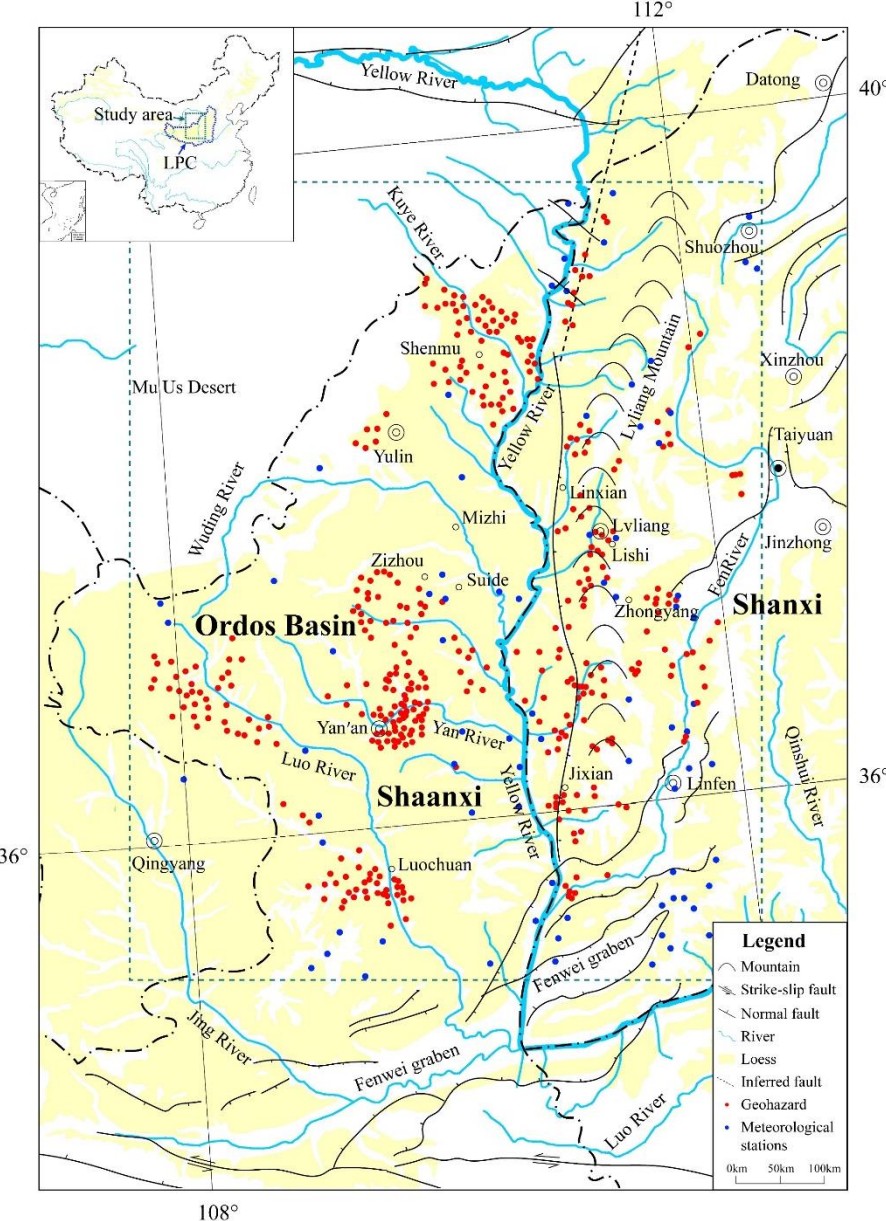

**Figure 2.** Geological map of the study area. The red dots denote the cracking–sliding failure cases, and the blue dots indicate the meteorological stations in the study area.

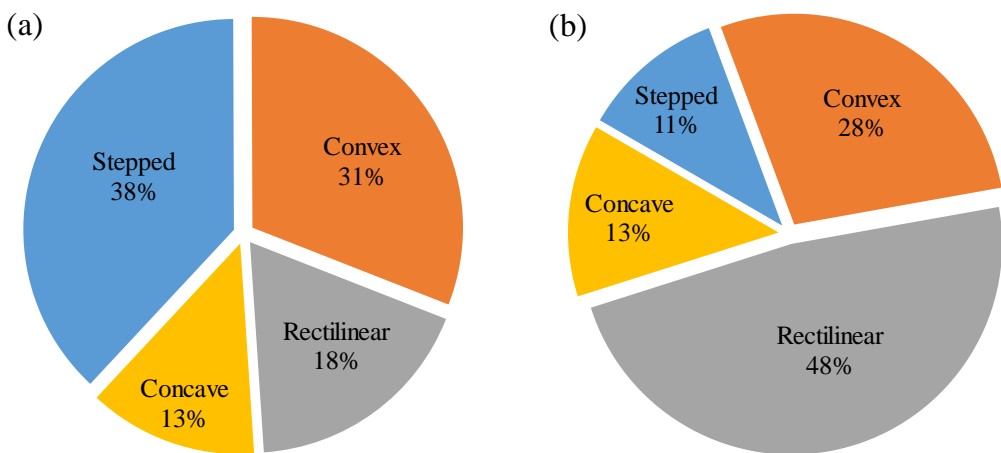

**Figure 3.** Statistical analysis results: a) classification of loess slopes in Lishi City, Shanxi Province, China, on the basis of a field survey of 212 loess slopes, indicating that stepped slopes are dominant in the study area. b) Percentage of cracking–sliding failures that occurred in different types of loess slopes across the study area, showing that rectilinear slopes are highly susceptible to loess failures.

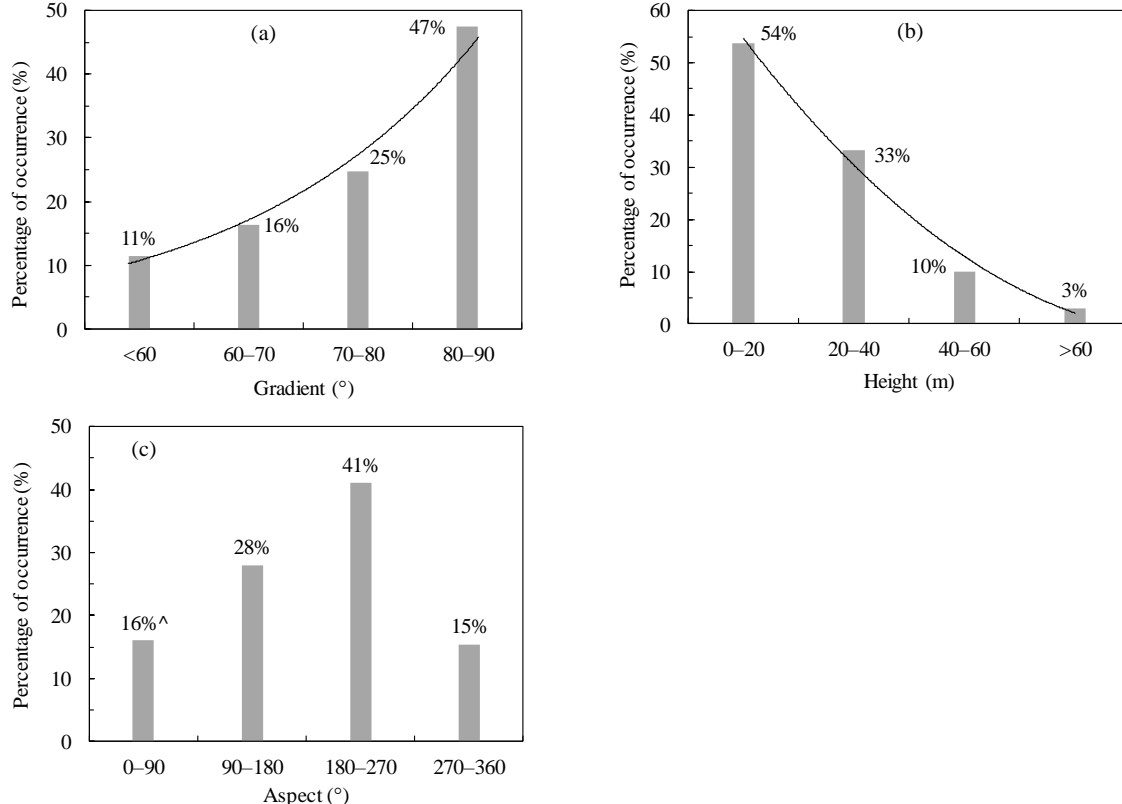

**Figure 4.** Effect of slope features on cracking–sliding failures.

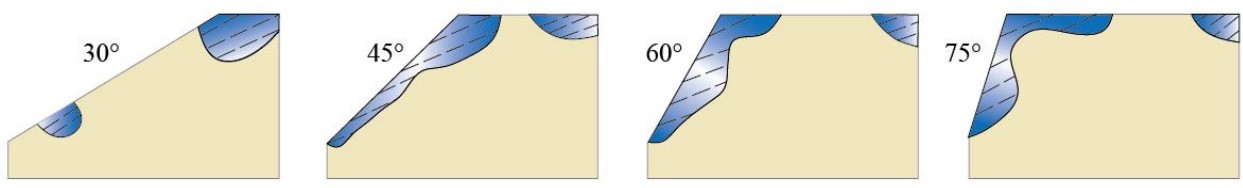

**Figure 5.** Development of tension zones in slopes of different gradients (Stacey, 1970; Zhang et al., 2009).

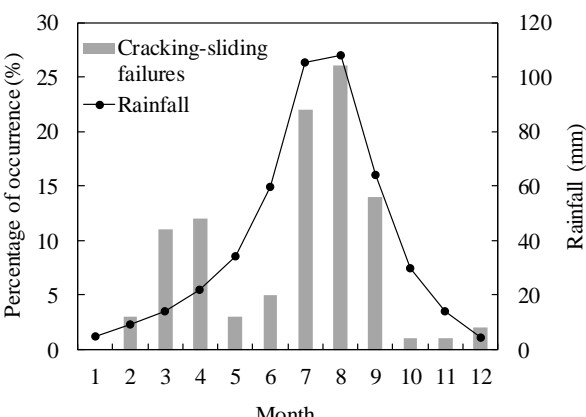

**Figure 6.** Occurrence of cracking–sliding failures mainly in July to September and consistent with the average monthly rainfall.

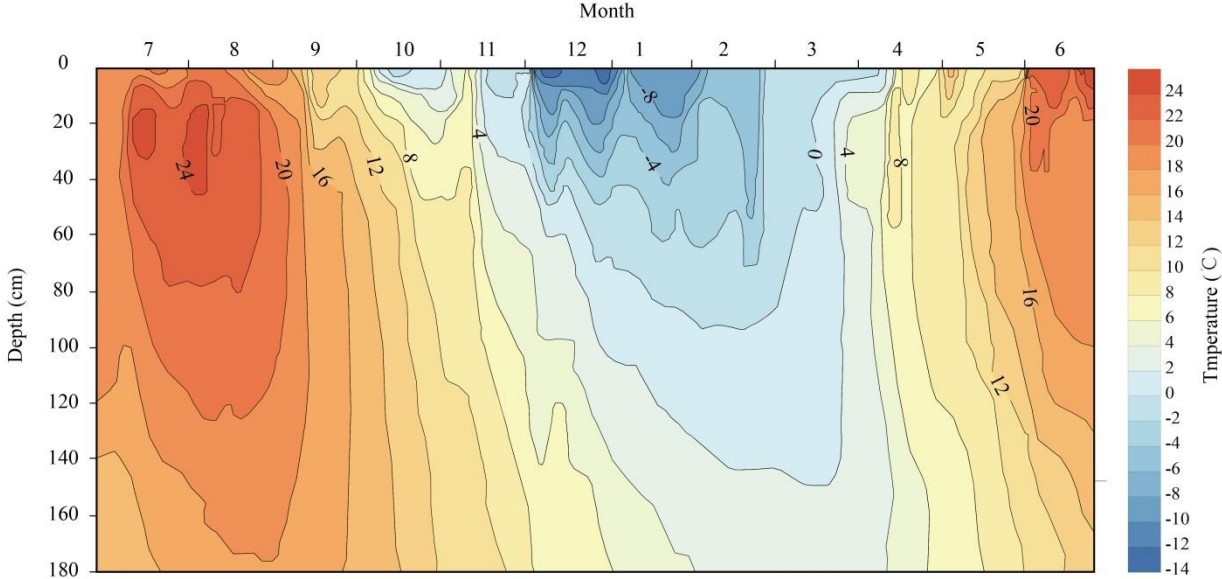

**Figure 7.** Annual variation of temperature (°C) within a shallow zone of a typical loess slope (Yang and Shao, 1995).

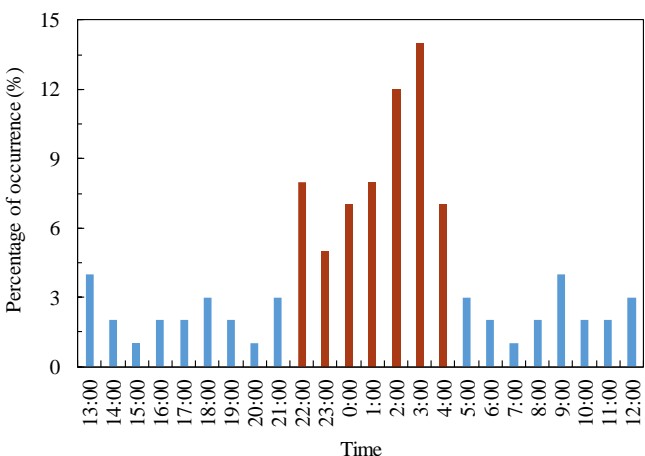

**Figure 8.** Temporal distribution of cracking–sliding failures in a day between 10 pm and 4 am.

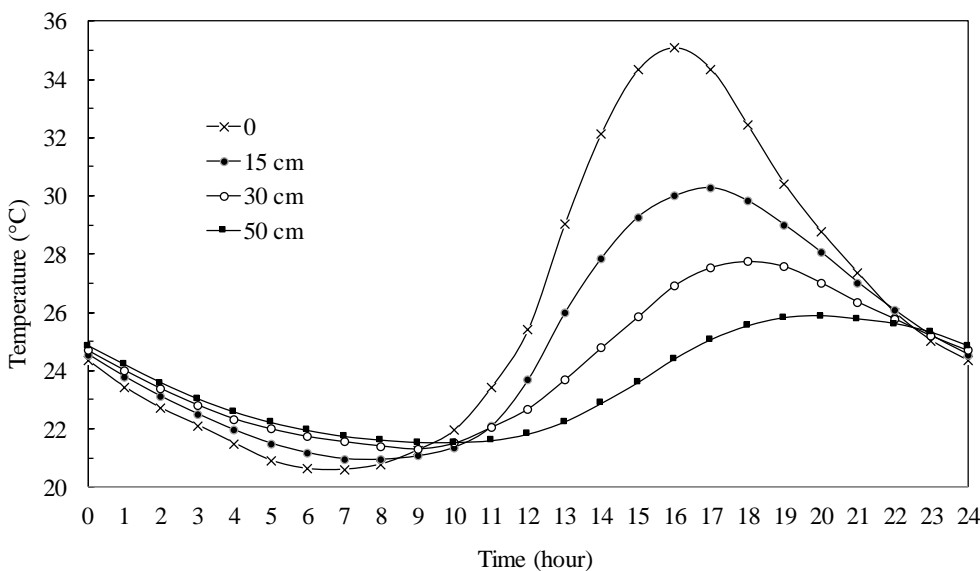

**Figure 9.** Daily soil temperature variation in loess areas of China in summer. Data are from the field monitoring during April 2014 to September 2017 in Linxian County, Shanxi, China.

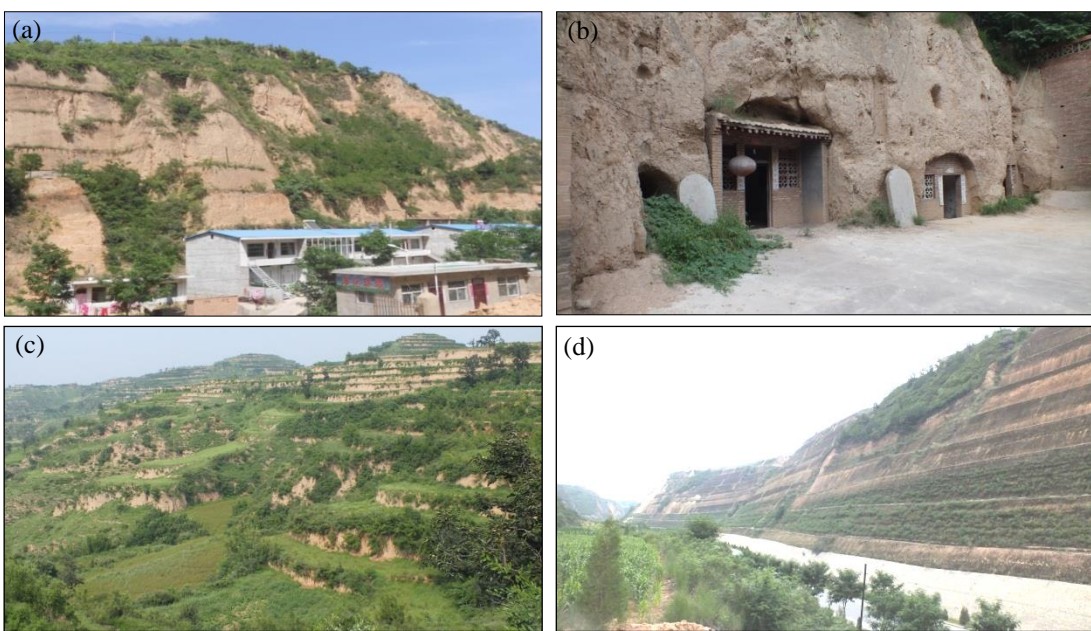

**Figure 10.** Typical engineering activities in loess areas in China: a) cut slopes for buildings; b) excavations for cave dwellings; c) terraced fields for farming; and d) cut slopes for highways.

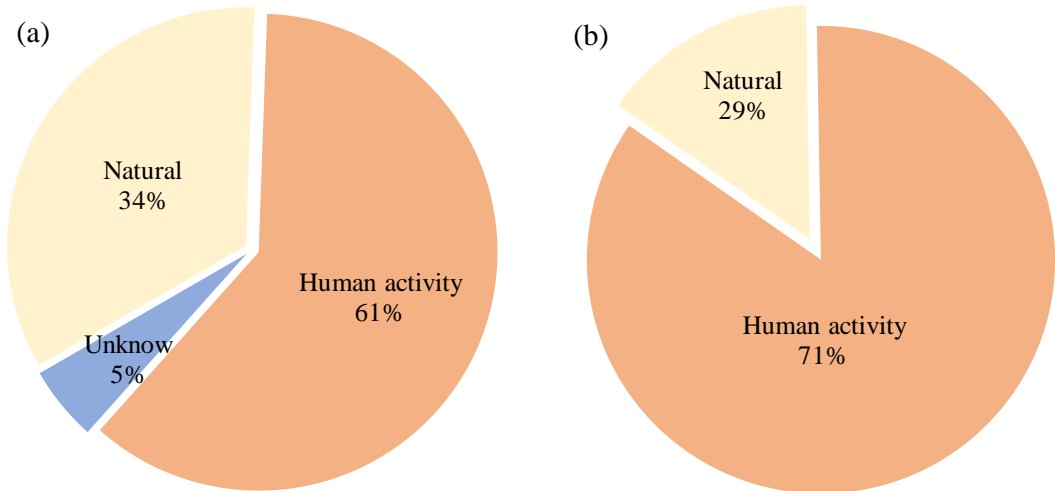

**Figure 11.** Role of engineering activities in loess failures: a) Shanxi Province; and b) Huangling County, Shaanxi Province.