# Peer review of "Factors influencing development of cracking-sliding failures of loess across the eastern Loess Plateau of China"

_Natural Hazards and Earth System Sciences, 2016_

## Referee Comment (RC1) · Anonymous Referee #1 · 20 Jan 2017

I have been nominated to act as reviewer of the manuscript "Influencing factors and development patterns of cracking–sliding failure of loess in China" by Jiarui Mao, Xiqiong Xiang, and Yanrong Li. The manuscript aims to characterize the landslides occurring on loess in a quite vast area of China, with the ambitious target of (i) preventing landslides occurrence, (ii) reduce losses due to this type of failures. Unfortunately, the paper appears to be not scientifically sound, rather simple in its explanation of phenomena, and data and analyses appear inadequate to fulfill the proposed targets mentioned above. Therefore, the paper is not ready for publication in Natural Hazards and Earth System Science. In the following, I provide some more specific details:

1. Research is poorly described, as well as the available datasets. No description on how data were collected has been provided, nor the data types (landslides are points or polygons, vector or raster?). As it is, the paper does not contain either novel data or new ideas/insights based on data collected by previous researchers. The only statistical analyses applied to data presented consist in a percentage comparison, which is rather simplistic compared to the many possible approaches applicable to analyse the influence of a set of variables on landslides occurrence, which is evident from the specific literature worldwide.

2. It is not clear to me where do the landslide data come from, what is their spatial, temporal and thematic accuracy, and, above all, their degree of homogeneity within all these accuracies. It is not clear what type of inventory is used (event, archive…) and which sources were used to get data. Even though the information comes from scientific literature, the authors appear not to fully handle the possible inhomogeneity of their data, and, therefore, the quality of the dataset.

3. The study area is $6.4 \times 10^5$ km$^2$, twice the surface of Italy, for example. In a landslide prone area such as the loess plateau, I am expecting a much larger dataset than a few hundreds (I take this number from a rough estimation of red dots in Figure 2). In such large areas, it is just not correct to simply draw some considerations from a small subsample of the landslides data available and extending them to areas where climatic, morphologic, anthropic, geologic conditions are (sometimes sensibly) different. In other words, to what extent the authors can extend to more than half a million square kilometre area the monthly ground temperature variations (Figure 8), or the daily temperature variations? For instance, daily temperature variations are considered for a small sample of 32 landslides. It is not stated where the sample is located, or if that location may be thought representative of all the Loess plateau. The same applies for the rainfall induced events. Comparing plots of figure 7, it is not clear why the Shaanxi has so many more failures of the other places. Is it just a larger area or it is an expression, for example, of the climatic variability inside the Loess plateau? I would have liked that the authors had pointed that out, and properly commented. If any, what areas can be represented by the Shaanxi? Why is it so different from the others in terms of rainfall thresholds? In the slope classification (Secion 2), again data are taken by a local study of Shaanxi and then applied to the whole Loess plateau. To what extent is it representative of an area that crosses two climatic zones?

Furthermore, also the temporal factor is important. For instance, temperature data come from observations that were obtained from a monitoring activity that last one year (from Nov 2004 to Oct 2005). Authors do not mention or deal with possible problems of temperature trends. Was it an average

year, or it was an exceptionally cold/hot one? If available, a plot of the average temperature of the last (or even following) 10 years would have helped.

4. Geography of area is not characterized. Elevations, slopes, aspects, hypsometry, should be statistically described. A little geomorphology and geology is described, but I expect something more clear and structured. There is no section on study area, with subsections dealing with geography, geomorphology, climate, and geology. In such broad studies, a reader expects these sections to be a frame to the analyses.

5. Language is often too generic. For example, words like "large dataset", "poor stability", "fair stability", "good stability" shouldn't be used in academic English of scientific papers. I also disagree with the term "cracking-sliding" as a type of failure. None of the existing landslides classifications encompass that term. Authors should find terms compliant with published nomenclature. If they are proposing a new nomenclature term, they should say that explicitly, and justify it in detail, it should be a totally different paper.

6. The manuscript does not follow a structure accepted for a scientific paper. (i.e. Introduction, study area and available data, methods, results, discussion, conclusion). In particular, I disagree with the idea of not writing a Discussion section. Instead, the authors have chosen to add very simple considerations while presenting some (unclear) data, which is scientifically questionable. Furthermore, the factors hypothesized to be influent for the development of "cracking-sliding" failures are presented singularly, therefore the interaction of these factors, and their specific role or possible chains of processes inducing landsliding remains hidden and unclear.

7. References are only from China, whereas loess research is not only produced in China, but also in all the other countries where loess deposit is present (as the authors point out).

I recommend rejection. If the authors wanted to re-submit the paper, they should address all the above mentioned issues, provide new insights based on the data/evidences available, rewrite the paper describing a clearly reproducible research, based also on international scientific literature. Furthermore, I would suggest that the authors focus on much smaller test areas, where they have good quality datasets on different variables, and where they can possibly carry out a much more detailed landslide inventory, also using satellite or aerial images. This approach would enlarge the time window of the observations. The test areas could even sum up to just a few hundreds of square kilometres, but should be accurately chosen to be representative of larger morpho-litho-climatic sectors. The advantage would be to handle more easily the data and draw well data-supported hypotheses.

Best regards.

---

## Referee Comment (RC2) · Anonymous Referee #2 · 29 Jan 2017

This manuscript presents an extensive list of factors and patterns of failures that occur in loess across the Chinese loess plateau. The manuscript is well organized and highly legible. I recommend its publication as it provides a rare and an effective summary of a frequent geohazard over a very large area, affecting lives of a significant human population.

---

## Author Comment (AC1) · 19 Feb 2017

We thank the reviewer very much for his comments. Please find below the responses to each point of the comments.

1. Research is poorly described, as well as the available datasets. No description on how data were collected has been provided, nor the data types (landslides are points or polygons, vector or raster?). As it is, the paper does not contain either novel data or new ideas/insights based on data collected by previous researchers. The only statistical analyses applied to data presented consist in a percentage comparison, which is rather simplistic compared to the many possible approaches applicable to analyse the influence of a set of variables on landslides occurrence, which is evident from the specific literature worldwide.

Response:

The paper is based on a painstaking review of numerous unpublished reports presented to the local governments and published literature. The factors influencing loess failures in the Loess Plateau of China are determined from the selected failure cases. Though the discussion utilizes statistics of these factors, the main goal is to demonstrate the correlations between loess failures and slope features (slope shape, angle, height, etc.), daily time period (from 9 pm to 4 am the next day), annual time periods (July to September, March to May) and human activities.

The value of this paper is that it presents an overall picture of cracking-sliding failures of loess in China and the mysteries (e.g., the dependence of loess failure occurrence on daily time periods) which demand further academic research.

Description of data types, sources, collection and validation will be added in the revision. This is further explained in our response to the next comment below.

2. It is not clear to me where do the landslide data come from, what is their spatial, temporal and thematic accuracy, and, above all, their degree of homogeneity within all these accuracies. It is not clear what type of inventory is used (event, archive...) and which sources were used to get data. Even though the information comes from scientific literature, the authors appear not to fully handle the possible inhomogeneity of their data, and, therefore, the quality of the dataset.

Response:

The dataset is from two sources: the events recorded by the government office and the events reported in publications. Both types are originally collected through field surveys, which are normally carried out within 1 to 2 days after each event. There are standard procedures to maintain the accuracy and reliability and therefore the quality of the dataset. The data cited in the scientific literature were individually verified and sifted by referring to the original records in the relevant government offices.

A brief description of the origin and nature of the dataset can be added in the revision.

3. The study area is 6.4×105 km2, twice the surface of Italy, for example. In a landslide prone area such as the loess plateau, I am expecting a much larger dataset than a few hundreds (I take this number from a rough estimation of red dots in Figure 2). In such large areas, it is just not correct to simply draw some considerations from a small subsample of the landslides data available and extending them to areas where climatic, morphologic, anthropic, geologic conditions are (sometimes sensibly) different. In other words, to what extent the authors can extend to more than half a million square kilometre area the monthly ground temperature variations (Figure 8), or the daily temperature variations? For instance, daily temperature variations are considered for a small sample of 32 landslides. It is not stated where the sample is located, or if that location may be thought representative of all the Loess plateau. The same applies for the rainfall induced events. Comparing plots of figure 7, it is not clear why the Shaanxi has so many more failures of the other places. Is it just a larger area or it is an expression, for example, of the climatic variability inside the Loess plateau? I would have liked that the authors had pointed that out, and properly commented. If any, what areas can be represented by the Shaanxi? Why is it so different from the others in terms of rainfall thresholds? In the slope classification (Secion 2), again data are taken by a local study of Shaanxi and then applied to the whole Loess plateau. To what extent is it representative of an area that crosses two climatic zones? Furthermore, also the temporal factor is important. For instance, temperature data come from observations that were obtained from a monitoring activity that last one year (from Nov 2004 to Oct 2005). Authors do not mention or deal with possible problems of temperature trends. Was it an average year, or it was an exceptionally cold/hot one? If available, a plot of the average temperature of the last (or even following) 10 years would have helped.

Response:

The events included in the dataset are only those that have caused deaths and economic losses, as only these events are surveyed and recorded. The red dots in Figure

2 are the events recorded and validated in the past 12 years, which account for a few hundreds. The Shaanxi Province has more failures than other places in Figure 7 because this province has denser population and therefore more recorded events.

We would like to take Shaanxi Province as the test area where climatic, morphologic, anthropic and geologic conditions are homogeneous. The temperature trends are from an average year. A plot of the average temperature of the last 10 years can be added in the revision, though there is no obvious difference in terms of temperature fluctuation from the year Nov 2004 to Oct 2005.

4. Geography of area is not characterized. Elevations, slopes, aspects, hypsometry, should be statistically described. A little geomorphology and geology is described, but I expect something more clear and structured. There is no section on study area, with subsections dealing with geography, geomorphology, climate, and geology. In such broad studies, a reader expects these sections to be a frame to the analyses.

Response:

A section on study area can be added in the revision with elevations, slopes, aspects etc. It should be noted that the cracking-sliding failures are developed within the loess layer. Furthermore, the loess is uniform in color, mineral and chemical composition throughout the Loess Plateau and does not show any relation to the local bedrock. It overlays bedrock and continuously covers basins, slopes, hills, valleys and terraces, making the present-day topographic relief consistent with the underlying terrain. This means that there is an identical geomorphology and geology over a broad area. This is also why we do not take geology as a factor to analyze.

5. Language is often too generic. For example, words like “large dataset”, “poor stability”, “fair stability”, “good stability” shouldn’t be used in academic English of scientific papers. I also disagree with the term “cracking-sliding” as a type of failure. None of the existing landslides classifications encompass that term. Authors should find terms compliant with published nomenclature. If they are proposing a new nomenclature

term, they should say that explicitly, and justify it in detail, it should be a totally different paper.

Response:

The mass movements in loess area of China are frequent and varied in forms of toppling, falling, cracking-sliding, sliding, peeling and caving. Among these modes, the cracking-sliding failure is the most common type at volumes of the order of 100 m3. The following explanation for the cracking-sliding failure mode is proposed and will be involved in the revision:

Unlike “flows” or “slides” as defined by Cruden and Varnes (1996), the cracking-sliding failures have composite failure planes, which are composed of two parts. The upper part normally develops vertically from the crown of the slope down to one to several meters deep. It is formed by tensile cracking and a slope can stand for a long time with such cracks before it fails. The lower part is generally inclined at an angle ranging from 15 to 60 degrees. The sliding, triggered by rainfall, freezing-thawing, daily temperature fluctuation, slope undercutting and earth tremors, along the lower part is thought to mobilize cracking-sliding failures. More than 1000 cracking-sliding failures were recorded in the past two decades caused on average more than 100 fatalities per year, despite small volumes of individual failures.

6. The manuscript does not follow a structure accepted for a scientific paper. (i.e. Introduction, study area and available data, methods, results, discussion, conclusion). In particular, I disagree with the idea of not writing a Discussion section. Instead, the authors have chosen to add very simple considerations while presenting some (unclear) data, which is scientifically questionable. Furthermore, the factors hypothesized to be influent for the development of “cracking-sliding” failures are presented singularly, therefore the interaction of these factors, and their specific role or possible chains of processes inducing landsliding remains hidden and unclear.

Response:

We think that a scientific paper can be structured in different ways as long as it gives substantial information and logic and reasonable discussions. As mentioned before, a section will be added to present the study area and dataset. Explanations and discussions will be revised according to the study area limited to the Shaanxi province. The interaction of influential factors cannot be determined from the existing dataset and is out of the scope of this paper.

7. References are only from China, whereas loess research is not only produced in China, but also in all the other countries where loess deposit is present (as the authors point out).

Response:

In the revision, we will include most relevant international publications.

8. I recommend rejection. If the authors wanted to re-submit the paper, they should address all the above mentioned issues, provide new insights based on the data/evidences available, rewrite the paper describing a clearly reproducible research, based also on international scientific literature. Furthermore, I would suggest that the authors focus on much smaller test areas, where they have good quality datasets on different variables, and where they can possibly carry out a much more detailed landslide inventory, also using satellite or aerial images. This approach would enlarge the time window of the observations. The test areas could even sum up to just a few hundreds of square kilometres, but should be accurately chosen to be representative of larger morpho-litho-climatic sectors. The advantage would be to handle more easily the data and draw well data-supported hypotheses.

Response:

We thank the reviewer for his valuable comments. It is impossible however to produce an overall picture of loess failures with every detail in one paper. We can address the comments on the origin of database and study area description, which seem to be the

main points of the reviewer's comments. The second reviewer strongly encourages us as he recognizes the scientific contribution and writing style of the manuscript.

In summary, we would like to revise the manuscript to the comments from Reviewer 1 as follows: 1. The study area: we agree to take Shaanxi Province as the study area. 2. The context: we agree to add necessary description and discussion about the study area, the database and the influencing factors. . 3. The database: we will present the setting of the database with more detailed description of the data sources as well as the data validation. 4. The failure mode: we will put more description and discussion about the cracking-sliding failure mode.

---

## Author Comment (AC2) · 19 Feb 2017

We thank the reviewer very much for his recognition of the scientific contribution and writing style of the manuscript. We will pursue publication of it.
* * *

---

## Author Response (AR1)

The authors thank the reviewer and editor very much for their encouraging and constructive comments. All comments have been addressed in this revision.

1. Research is poorly described, as well as the available datasets. No description on how data were collected has been provided, nor the data types (landslides are points or polygons, vector or raster?). As it is, the paper does not contain either novel data or new ideas/insights based on data collected by previous researchers. The only statistical analyses applied to data presented consist in a percentage comparison, which is rather simplistic compared to the many possible approaches applicable to analyse the influence of a set of variables on landslides occurrence, which is evident from the specific literature worldwide.

Response:
    The paper is based on a painstaking review of numerous unpublished reports presented to the local governments and published literature. The factors influencing loess failures in the Loess Plateau of China are determined from the selected failure cases. Though the discussion utilizes statistics of these factors, the main goal is achieved through demonstrating the correlations between loess failures and slope features (slope shape, angle, height, etc.), daily time period (from 10 pm to 4 am the next day), annual time periods (July to September, March to April) and human activities.
    Description of data types, sources, collection and validation have been added in the revision. This is further explained in our response to the next comment below.

2. It is not clear to me where do the landslide data come from, what is their spatial, temporal and thematic accuracy, and, above all, their degree of homogeneity within all these accuracies. It is not clear what type of inventory is used (event, archive…) and which sources were used to get data. Even though the information comes from scientific literature, the authors appear not to fully handle the possible inhomogeneity of their data, and, therefore, the quality of the dataset.

Response:
    The dataset is from two sources: the events recorded by the government office and the events reported in publications. Both types were originally collected through field surveys, which were normally conducted within 1 to 2 days after each event. There are standard procedures to maintain the accuracy and reliability and therefore the quality of the dataset. The data cited in the scientific literature were individually verified and sifted by referring to the original records in the relevant government offices.
    A description of the origin and nature of the dataset has been added in the revision. Please find it in **Section 3 Dataset**.

3. The study area is 6.4×105 km2, twice the surface of Italy, for example. In a landslide prone area such as the loess plateau, I am expecting a much larger dataset than a few hundreds (I take this number from a rough estimation of red dots in Figure 2). In such large areas, it is just not correct to simply draw some considerations from a small subsample of the landslides data available and extending them to areas where climatic, morphologic, anthropic, geologic conditions are (sometimes sensibly) different. In other words, to what extent the authors can extend to more than half a million square kilometre area the monthly ground temperature variations (Figure 8), or the daily temperature variations? For instance, daily temperature variations are considered for a small sample of 32 landslides. It is not stated where the sample is located, or if that location may be thought representative of all the Loess plateau. The same applies for the rainfall induced events. Comparing plots of figure 7, it is not clear why the Shaanxi has so many more failures of the other places. Is it just a larger area or it is an expression, for example, of the climatic variability inside the Loess plateau? I would have liked that the authors had pointed that out, and properly commented. If any, what areas can be represented by the Shaanxi? Why is it so different from the others in terms of rainfall thresholds? In the slope classification (Secion 2), again data are taken by a local study of Shaanxi and then applied to the whole Loess plateau. To what extent is it representative of an area that crosses two climatic zones?

Furthermore, also the temporal factor is important. For instance, temperature data come from observations that were obtained from a monitoring activity that last one year (from Nov 2004 to Oct 2005). Authors do not mention or deal with possible problems of temperature trends. Was it an average year, or it was an exceptionally cold/hot one? If available, a plot of the average temperature of the last (or even following) 10 years would have helped.

Response:

We updated **Section 2** and **Section 3** according to the reviewer's comment.

The study area has been limited to the east of the LPC covering the regions of Northern Shaanxi and Western Shanxi provinces, considering their homogeneous background of climatic, morphologic, geologic, and anthropic conditions. Please refer to Figure 1 in the revision.

Data pertaining to rainfall are obtained from the records of 75 meteorological stations (blue 102 dots in Fig. 1), which are almost uniformly distributed across the study area. Statistical analysis shows that the variation in average annual rainfall in the past 15 years among these stations is less than 80 mm, indicating a relatively homogeneous climatic condition over the study area.

A description of the temperature data is added. As shown in Fig. 8, the average daily temperature of August is from a field monitoring from April 2014 to September 2017 in Linxian County, Shanxi, China.

4. Geography of area is not characterized. Elevations, slopes, aspects, hypsometry, should be statistically described. A little geomorphology and geology is described, but I expect something more clear and structured. There is no section on study area, with subsections dealing with geography, geomorphology, climate, and geology. In such broad studies, a reader expects these sections to be a frame to the analyses.

Response:

**Section 2: Study area** describes elevations, slopes, aspects etc in the study area. It should be noted that the cracking-sliding failures are developed within the loess layer. The loess is uniform in color, mineral and chemical composition throughout the Loess Plateau and does not show any relation to the local bedrock. It overlays bedrock and continuously covers basins, slopes, hills, valleys and terraces, making the present-day topographic relief consistent with the underlying terrain. This means that there is an identical geomorphology and geology over a broad area. This is also why we do not take geology as a factor to analyze.

5. Language is often too generic. For example, words like "large dataset", "poor stability", "fair stability", "good stability" shouldn't be used in academic English of scientific papers. I also disagree with the term "cracking-sliding" as a type of failure. None of the existing landslides classifications encompass that term. Authors should find terms compliant with published nomenclature. If they are proposing a new nomenclature term, they should say that explicitly, and justify it in detail, it should be a totally different paper.

Response:

The mass movements in loess area of China are frequent and varied in forms of toppling, falling, cracking-sliding, sliding, peeling and caving. Among these modes, the cracking-sliding failure is the most common type at volumes of the order of 100 m$^3$. The following explanation for the cracking-sliding failure mode has been involved in the revision:

Unlike "flows" or "slides" as defined by Cruden and Varnes (1996), the cracking-sliding failures have composite failure planes, which are composed of two parts. The upper part normally develops vertically from the crown of the slope down to one to several meters deep. It is formed by tensile cracking and a slope can stand for a long time with such cracks before it fails. The lower part is generally inclined at an angle ranging from 15 to 60 degrees. The sliding, triggered by rainfall, freezing-thawing, daily temperature fluctuation, slope undercutting and earth tremors, along the lower part is thought to mobilize cracking-sliding failures. More than 1000 cracking-sliding failures were recorded in the past two decades caused on average more than 100 fatalities per year, despite small volumes of individual failures.

6. The manuscript does not follow a structure accepted for a scientific paper. (i.e. Introduction, study area and available data, methods, results, discussion, conclusion). In particular, I disagree with the idea of not writing a Discussion section. Instead, the authors have chosen to add very simple considerations while presenting some (unclear) data, which is scientifically questionable. Furthermore, the factors hypothesized to be influent for the development of "cracking-sliding" failures are presented singularly, therefore the interaction of these factors, and their specific role or possible chains of processes inducing landsliding remains hidden and unclear.

Response:
As suggested, sections were added to present the study area and dataset. Explanations and discussions was revised accordingly.
The interaction of influential factors cannot be determined from the existing dataset and is out of the scope of this paper.

7. References are only from China, whereas loess research is not only produced in China, but also in all the other countries where loess deposit is present (as the authors point out).

Response:
We referred to and included the most relevant international publications, e.g., Smalley et al., 2011; Stacey, 1970; and Sprafke and Obreht, 2016.

[revised manuscript text omitted]

---

## Author Response (AR2)

Manuscript: nhess-2016-345-manuscript-version4

Title: Factors influencing development of cracking-sliding failures of loess across the east Loess Plateau of China

| Reviewer #1 | |
|---|---|
| **Comments** | **Responses** |
| 1. Regarding the title, "development pattern" seems a little vague. The authors may try "Factors influencing development of cracking–sliding failures in loess across the east of the Loess Plateau in China" | Revised as suggested. |
| 2. Suggest to check the formatting of the whole paper to meet the requirements of the journal, especially, the author list, citations in the text and reference list. | As suggested, we have checked and revised the formatting of the whole paper. |

| Reviewer #2 | |
|---|---|
| **Comments** | **Responses** |
| 1. The paper is very interesting because it reports an example of a not very common typology of failure occurring in Loess Plateau of China. The manuscript is well organized, all figures and tables included in the text are necessary and appropriate. Nevertheless, some references are missed in the references section (e.g. Li and Shi, 2017; Li and Mo, 2017). | Noticed with thanks and addressed as suggested. |
| 2. Cracking-sliding failures are the most common type of landslide in Loess Plateau of China, but they are not very known in the scientific literature | As suggested, we have added a photo of typical cracking-sliding failure of loess (Fig. 1), which occurred in Shilou County of Shanxi Province recently on March 10, |

| | |
|---|---|
| and they are not encompassed in the standard classification of landslides. Could the authors add some photos in the paper? | 2018, destroying 36 houses. Detailed description is also given in Line 55. |
| 3. With respect to the previous version of the paper the authors added the description of the data type, sources, collection and validation, but the typologies of data that the authors have used are not yet very clear: a) how are landslides represented (polygons or points)? b) in order to perform the statistical analysis the authors have compared, with the other factors, the whole landslide or only the initiation area? c) which data (e.g. type of DEM and its resolution) were used in order to obtain slope profile, gradient, height and aspect? | Slope profile, gradient, height and aspect, were derived in polygon from the initiation areas. When carrying out the comparison the initiation areas rather than the whole landslides were compared. The polygons were obtained by means of 1) interpretation of remote sensing images which were taken prior to the event; 2) engineering drawings if the host slope was engineered; or 3) post-event field survey and consultation with the local populace.

The above explanation is incorporated in Line 98 in the text. |

We thank the editor and reviewers very much for providing valuable comments and suggestions. By addressing these comments, the quality of the paper got further improvement.

---

## Author Response (AR3)

Manuscript: nhess-2016-345

Title: Factors influencing development of cracking-sliding failures of loess across the eastern Loess Plateau of China

| Comments | Responses |
|---|---|
| What do you mean? Geohazard, or what? | It's geohazard. We have revised it in the manuscript accordingly. |
| I do not understand this. Do you mean slopes mostly exposed to the sun light? Please rewrite in better English. | Yes. Addressed as suggested. |
| I believe there are more older citations of this for the term loess. You should probably indicate the oldest ones, followed by this one, if you like. | The oldest citations are referenced. Richthofen, 1882 & Pye, 1987 |
| Please specify whether you are here referring to failure of natural karst cave, or of cavity built by man. You could refer to the following articles: Gutierrez F., Parise M., De Waele J. & Jourde H., 2014, A review on natural and human-induced geohazards and impacts in karst. Earth Science Reviews, vol. 138, p. 61-88, doi: 10.1016/j.earscirev.2014.08.002. Parise M., 2013, Artificial caves as a possible danger: sinkholes and other effects at the surface. Opera Ipogea, n. 1, p. 95-102. Parise M., 2015, A procedure for evaluating the susceptibility to natural and anthropogenic sinkholes. Georisk, vol. 9 (4), p. 272-285, DOI:10.1080/17499518.2015.10 45002. | The caving in the manuscript refers to the failure of cavity built by human. This is clarified in the revised manuscript. |
| Poor english, please rewrite. | The language is reconstructed as suggested. Thanks. |

| | |
|---|---|
| I do not understand. Is subsidence due to tectonics, or to gravitational processes? Please explain better. | The subsidence is due to tectonic movements. It is clarified in the revision. |
| On which bases did you exclude the others? What were the criteria you used for select the sample to analyze? | The samples were selected by verifying the reliability, accuracy, and completeness of the original records of the failure events. |
| I would change this term in something like "Slopes mostly exposed to sunlight" | Addressed as suggested. |
| It would be interesting to know if you have the time of failure for all the cases you selected, or not | Yes, we have the time of failure for all selected cases and the analysis are based on these records. |
| You should check also this paper:

Del Prete S. & Parise M., 2013, An overview of the geological and morphological constraints in the excavation of artificial cavities. In: Filippi M. & Bosak P. (Editors), Proceedings 16th International Congress of Speleology, Brno, 21-28 July 2013, vol. 2, p. 236-241. | Thanks for the recommendation. |
| Colors are too dark, writing is not clear | Addressed as suggested. |

**All comments from the editor have been addressed in this revision. Thank the editor very much for these valuable suggestions and corrections.**